# Switchable Token-Specific Codebook Quantization For Face Image Compression

**Yongbo Wang**[*]
East China Normal University
Shanghai, China

**Haonan Wang**[*]
Tencent Youtu Lab
Shanghai, China

**Guodong Mu**
Tencent Youtu Lab
Shanghai, China

**Ruixin Zhang**
Tencent Youtu Lab
Shanghai, China

**Jiaqi Chen**
East China Normal University
Shanghai, China

**Jingyun Zhang**
Tencent WeChat Pay Lab33
Shenzhen, China

**Jun Wang**
Tencent WeChat Pay Lab33
Shenzhen, China

**Yuan Xie**
East China Normal University
Shanghai, China

**Zhizhong Zhang**[†]
East China Normal University
Shanghai, China

**Shouhong Ding**
Tencent Youtu Lab
Shanghai, China

{51265901105,51275901135}@stu.ecnu.edu.cn, {yxie,zzzhang}@cs.ecnu.edu.cn
{quinnhnwang,gordonmu,ruixinzhang,naskyzhang,earljwang,ericshding}@tencent.com

## Abstract

With the ever-increasing volume of visual data, the efficient and lossless transmission, along with its subsequent interpretation and understanding, has become a critical bottleneck in modern information systems. The emerged codebook-based solution utilize a globally shared codebook to quantize and dequantize each token, controlling the bpp by adjusting the number of tokens or the codebook size. However, for facial images—which are rich in attributes—such global codebook strategies overlook both the category-specific correlations within images and the semantic differences among tokens, resulting in suboptimal performance, especially at low bpp. Motivated by these observations, we propose a Switchable Token-Specific Codebook Quantization for face image compression , which learns distinct codebook groups for different image categories and assigns an independent codebook to each token. By recording the codebook group to which each token belongs with a small number of bits, our method can reduce the loss incurred when decreasing the size of each codebook group. This enables a larger total number of codebooks under a lower overall bpp, thereby enhancing the expressive capability and improving reconstruction performance. Owing to its generalizable design, our method can be integrated into any existing codebook-based representation learning approach and has demonstrated its effectiveness on face recognition datasets, achieving an average accuracy of 93.51% for reconstructed images at 0.05 bpp.

---

[*]Equal contribution. This work was done by Yongbo Wang during an internship at Tencent Youtu Lab.
[†]Corresponding author.

39th Conference on Neural Information Processing Systems (NeurIPS 2025).

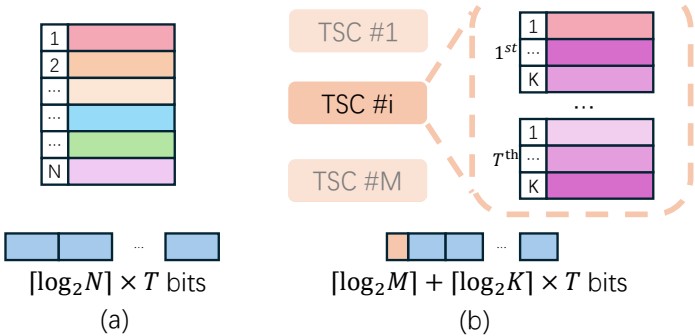

Figure 1: Storage cost comparison between previous latent space models and our method. (a) Previous latent space model: Global-shared codebook requiring storage cost of $T\lceil\log_2 N\rceil$ bits. (b) Our method: Token-specific codebook selection reduces storage to $T\lceil\log_2 K\rceil + \lceil\log_2 M\rceil$ bits.

# 1  Introduction

The volume of image data produced daily by smart device has skyrocketed. However, due to bandwidth limitations and storage costs, such data are typically stored and transferred in lossy compressed formats instead of raw RGB data. While lossy compression can greatly reduce storage requirements, it inevitably leads to a drop in visual quality and significantly impairs the performance of certain machine perception tasks, such as face recognition (1; 2).

In recent years, numerous solutions have been proposed with the aim of achieving high-fidelity image reconstruction and maintaining recognition capabilities with extremely low bpp. For instance, traditional compression techniques such as JPEG (3), GIF (4), and WebP (5) remain widely adopted due to their strong compatibility to trade off computational complexity, storage size, and visual quality. Meanwhile, with advancements in deep learning, neural network-based compression methods have gained increasing traction. VQ-VAE (6) transforms images into discrete indices mapped to a codebook, enabling the compression of 2D pixel spaces into compact latent spaces. Despite their expressive feature representation, these methods are constrained by the requirement to maintain 2D structural correspondence, which prevents them from fully leveraging spatial redundancy in images and limits further reduction of bpp. To address this limitation, TiTok (7) introduced a latent code framework that encodes images into a 1D latent space, achieving high-efficiency compression of $256 \times 256$ images with only 32 tokens.

However, when targeting even lower bpp, these methods primarily rely on reducing the number of tokens or decreasing the size of the codebook. Unfortunately, both strategies result in severe degradation of visual quality for compressed images and significant drops in machine recognition performance. Through a detailed analysis, we identify a pivotal bottleneck in existing VQ-VAE-style methods: all tokens share a single global codebook. To ensure that the codebook accommodates the diverse features of all images, it must be sufficiently large. Consequently, reducing the size of the codebook leads to a drastic performance decline. This raises an important question: **Can we reorganize the codebook to simplify the problem into smaller, more manageable subproblems?**

Taking face images as an example, variations in attributes such as gender, age, and ethnicity suggest that images with similar attributes often share similar features. Therefore, the global shared codebook can be replaced with multiple codebooks, each designed for images with specific attributes. By enabling images to selectively use a suitable codebook, the complexity of each codebook's task can be reduced. Furthermore, within a single image, regardless of the feature extraction architecture used (i.e., CNN or ViT), different tokens explicitly or implicitly represent semantic information pertaining to different aspects of the image. For example, some tokens may correspond to facial regions, while others may be associated with the image's category. Forcing all tokens to share the same codebook increases learning difficulty. To this end, we propose a token-specific codebook quantization mechanism, where each token is assigned its own unique sub-codebook, significantly reducing the capacity requirements of individual sub-codebooks.

Based on the above analysis, we propose a switchable token-specific codebook quantization mechanism that combines image-level and token-level segmentation and has been verified on multiple face datasets. In our method, a codebook routing module determines which codebook within the codebook pool is appropriate for a given image. Within the selected codebook, each token is assigned a sub-codebook tailored for its specific characteristics. In this way, the entire codebook pool can offer greater capacity, enabling improved compression and reconstruction performance. Additionally, as illustrated in Figure 1, due to our hierarchically dynamic structure, the actual storage overhead decreases from $T \times \lceil \log_2 N \rceil$ to $T \times \lceil \log_2 K \rceil + \lceil \log_2 M \rceil$, enabling stronger feature representations under the same or even lower bpp. This ultimately results in substantial performance improvements.

We summarize our main contributions as follows:

1. We first introduce a switchable codebook quantization mechanism. By adjusting the bit width of the routing module and the size of the codebooks, our method supports flexible bpp configurations and increases total codebook capacity under the approximation of bpp, thereby enhancing overall performance.

2. We analyze intra-image token characteristics and propose a token-specific codebook quantization mechanism , thereby reducing the complexity of each codebook and improving overall performance.

3. We propose a hierarchically dynamic codebook structure that incorporates both image-level and token-level codebook partitioning. This module is plug-and-play and can be seamlessly integrated with state-of-the-art codebook-based compression methods.

4. We evaluate our method on the face recognition task and demonstrate its effectiveness with extensive experiments. Compared to the state-of-the-art method (TiTok), our approach achieves higher accuracy at the same bpp (e.g., improving accuracy from 87.56% to 91.66% at 0.0234 bpp) or reduces bpp at the same accuracy (e.g., from 0.0234 to 0.0157 at 87% accuracy).

# 2 Related Works

## 2.1 Lossy Image Compression

Traditional lossy image compression frameworks typically employ manually crafted pipelines, as exemplified by standards such as JPEG (3), JPEG2000 (8), HEVC (9), and VVC (10). However, isolated module optimizations prevent partial improvements from translating into global performance gains, inherently limiting the evolvability of such frameworks. Building upon advances in neural networks, Ballé et al. (11) pioneered convolutional neural networks (CNN)-based end-to-end optimized nonlinear transform coding framework, in which the analysis/synthesis transforms and entropy models are jointly trained to outperform traditional codecs in rate-distortion performance, and further extended this approach using a variational autoencoder formulation (12). Subsequent studies have advanced neural image compression by exploring improvements in network architectures (13; 14; 15), quantization methods (16; 17), entropy modeling techniques (18; 19), and optimization objectives (20; 21).

Beyond neural image compression, Agustsson et al. (22) introduced generative image compression to address blurred reconstructions at low bitrates inherent in prior methods, leveraging perceptual loss optimization for realistic synthesis. While early generative image compression frameworks primarily employed generative adversarial networks (GANs) (22; 23; 24), recent work has explored text/sketch-guided diffusion models (25), non-binary discriminator with quantized conditioning (26), and VQ-VAE-based latent-space transform coding (27), achieving high-fidelity and high-realism reconstructions under ultra-low bitrate constraints (28; 29). In addition, some specialized image compression frameworks have emerged for domain-specific tasks. For facial image compression, studies (30; 31) investigate racial bias induced by lossy compression in face recognition. Others develop frameworks tailored for facial images by utilizing edge maps (32) or semantic priors (33). However, these approaches either struggle to achieve effective compression under ultra-low bitrate constraints, or overlook the critical role of identity information in facial recognition tasks, resulting in insufficient exploration of recognition accuracy and identity consistency.

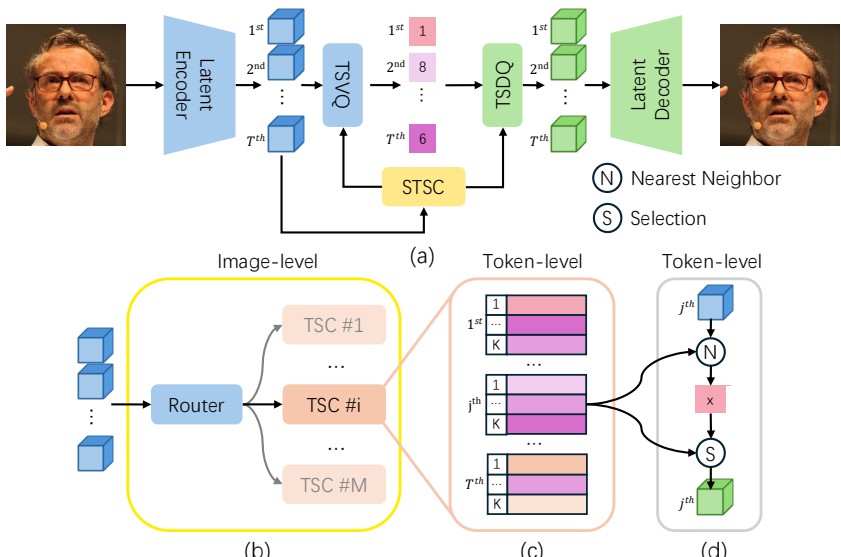

Figure 2: (a) Overview of the proposed architecture. (b) Dynamic switching mechanism for token-specific codebook selection. (c) Composition of the $i$-th token-specific codebook. (d) Token-specific quantization and dequantization process for the $j$-th token. Sample face images are from the FFHQ dataset (44).

## 2.2 Latent Space Model

Latent space model, initially developed for visual generation tasks, compresses high-dimensional raw pixels into compact latent representations for image synthesis. While variational autoencoders (VAEs) (34) map images into a continuous latent space, vector-quantized VAEs (VQ-VAEs) (6; 35) learn discrete latent representations via codebook learning, offering enhanced controllability and compression capability. Extending this framework, VQGAN (36) integrates perceptual loss and adversarial training to maintain high perceptual quality at elevated compression rates, further bridging generative modeling with discrete latent space compression. Recent advances have demonstrated the potential of latent space model: Rombach et al.(37) implement high-resolution image synthesis by performing diffusion processes in the latent space of VQGAN; Yu et al. (7) break away from conventional 2D latent grids by learning 1D token sequences for more flexible latent representations; and Shi et al. (38) pioneer scalable training paradigms to enable large-scale high-dimensional codebooks, significantly improving the utilization of large-scale codebooks.

Latent space modeling has also shown significant promise in facial processing tasks. Wang et al. (39)achieve high-fidelity and generalizable talking face generation by leveraging a pre-trained codebook to encode target faces. Tan et al. (40) further advance this domain by designing a unified codebook capable of representing diverse identities and expressions. Similarly, works (41; 42; 43) employ VQGAN for blind face restoration, using codebooks pretrained on high-quality facial images as priors to guide degraded image reconstruction. However, prior research predominantly focuses on spatial token compression within fixed codebook frameworks, where further reduction of token counts has reached diminishing returns. Our work pioneers a codebook-centric methodology by proposing a universal switchable token-specific codebook quantization. This innovation enhances compression efficiency through dynamic codebook specialization while maintaining reconstruction fidelity.

# 3 Method

## 3.1 Preliminary

The key idea of the latent space model lies in learning discrete latent representations to establish a compressed semantic space for image compression and generation. A typical latent space model comprises three fundamental components: an encoder $Enc$, a vector quantizer $Quant$, and a decoder

*Dec.* The encoder hierarchically compresses high-resolution images into compact latent vectors. The quantizer, rather important in compression, maintains an embedding codebook $\mathcal{C} \in \mathbb{R}^{N \times d}$ with $N$ learnable latent vectors that defines a discrete projection space. Through vector quantization, the continuous latent vectors produced by the encoder are discretized into codebook indices, thereby converting the image into a sequence of symbolic tokens. Specifically, given an input image $x \in \mathbb{R}^{H \times W \times 3}$, the corresponding discrete feature map can be formulated as follows:

$$\mathbf{z}_e = \text{Enc}(x), \tag{1}$$

$$\mathbf{z}_q = \underset{\mathcal{C}}{\text{Quant}}(\mathbf{z}_e) = \underset{\mathbf{e} \in \mathcal{C}}{\arg\min} \|\mathbf{z}_e - \mathbf{e}\|_2^2, \tag{2}$$

where $z_e, z_q \in \mathbb{R}^{h \times w \times d}$. The quantizer vias substituting continuous latent vectors with their nearest neighbors in a learnable codebook, thereby reformulating the input image as a compressed index sequence. This transformation achieves significant storage efficiency while preserving critical visual fidelity. For example, for the codebook $\mathcal{C} \in \mathbb{R}^{N \times d}$, each token requires only $\lceil \log_2 N \rceil$ bits. Thus, bpp is determined by both the codebook size and the token count.

The decoder operates on these discrete latent embeddings to reconstruct the input image $\hat{x} = \text{Dec}(\mathbf{z}_q)$. The training objective of the latent space model harmonizes three critical aspects: reconstructing error minimization, quantization error minimization, and perceptual quality preservation. The loss function can be formulated as:

$$\mathcal{L}_{\text{VQ}} = \|x - \hat{x}\|_2^2 + \|sg(\mathbf{z}_e) - \mathbf{z}_q\|_2^2 + \|sg(\mathbf{z}_q) - \mathbf{z}_e\|_2^2 + \lambda_p \mathcal{L}_{per}, \tag{3}$$

where $\mathcal{L}_{per}$ indicates the perceptual loss, and $sg(\cdot)$ refers to the stop-gradient operation. As illustrated in Figure 2, our method adheres to the latent space model but introduces a critical innovation: replacing conventional static codebooks with switchable token-specific codebooks, achieving superior rate-distortion performance compared to prior latent space models.

## 3.2 Switchable Codebook Quantization

The above analysis illustrates that the bpp is influenced by both the number of tokens and the bit-width of token indices. Since the number of tokens corresponds to the model architecture, a viable strategy is to reduce the bit-width of token indices. The bit-width of token indices $L$ is determined by the codebook size $N$, following the relation $L = \lceil \log_2 N \rceil$. However, simply decreasing the codebook size will negatively impact the reconstruction performance by limiting the diversity of codes available. Moreover, even halving the codebook size only reduces each index by one bit, which significantly diminishes the representational capacity of the latent space.

The inherent variations in facial attributes (e.g., gender, age, ethnicity) suggest that samples sharing common attributes exhibit analogous feature distributions. To mitigate the diminished codebook diversity caused by codebook compression, we propose Switchable Codebook Quantization (SCQ). Given an original codebook $\mathcal{C}_{\text{orig}} \in \mathbb{R}^{N \times d}$, we replace it with $M$ learnable codebooks $\{C^i \in \mathbb{R}^{\frac{N}{2^s} \times d}\}_{i=1}^M$, where $s \leq M$. This design ensures that code diversity remains comparable to or exceeds that of $\mathcal{C}_{\text{orig}}$ while allowing storage compression.

Specifically, the original latent space model requires $n \times b$ bits (for $n$ tokens with $b$-bit width), whereas SCQ reduces per-token bit-width by $s$ and introduces only $\log_2 M$ additional bits to switch codebook. The multiplicative reduction in bit-width ($\propto n(b-s)$) dominates the additive overhead ($+ \log_2 M$). For instance, when an image is represented by 256 tokens, replacing the original 4096-entry codebook with 256 codebooks (each containing 256 entries) reduces total bit allocation from 3072 bits to 2056 bits, achieving a 33% reduction in bpp.

During quantization, as shown in Figure 2(b) each image selects a codebook via a router $G$ that maps encoded features $\mathbf{z}_e$ to their optimal codebook partition. The selected codebook then quantizes $\mathbf{z}_e$ into discrete indices through the nearest-neighbor search, preserving constrained bpp.

$$k = G(\mathbf{z}_e), \tag{4}$$

$$\mathbf{z}_q = \underset{\mathcal{C}^i}{\text{Quant}}(\mathbf{z}_e) = \underset{\mathbf{e} \in \mathcal{C}^i}{\arg\min} \|\mathbf{z}_e - \mathbf{e}\|_2^2. \tag{5}$$

## 3.3 Codebook Routing

Increasing the number of codebooks introduces a corresponding selection challenge, as it becomes necessary to determine which codebook is most appropriate for a given input or context. To minimize quantization error, the most straightforward routing strategy is to compute quantization errors across all codebooks and select the one with minimal error:

$$G_{\text{naive}}(\mathbf{z}_e) = \underset{i \in \{1, \ldots, M\}}{\arg\min} \, \|\mathbf{z}_e - \underset{\mathcal{C}^i}{\text{Quant}}(\mathbf{z}_e)\|_2^2. \tag{6}$$

However, direct error-based routing may lead to preferential optimization of better-optimized codebooks, thereby compromising codebook diversity. Inspired by Mixture-of-Experts (45; 46) (MoE), we design a differentiable routing network $G_\theta$ composed of probabilistic sub-routers:

$$G_\theta(\mathbf{z}_e) = \underset{i \in \{1, \ldots, M\}}{\arg\max} \, g_\theta^i(\mathbf{z}_e), \tag{7}$$

where $g_\theta^i$ computes the selection probability for the $i$-th codebook. To ensure full codebook utilization, we introduce three loss functions:

$$\mathcal{L}_{\text{ent}} = \sum_{i=1}^{M} \bar{g}_\theta(\mathbf{z}_e) \log \bar{g}_\theta(\mathbf{z}_e), \tag{8}$$

$$\mathcal{L}_{\text{dec}} = -\frac{1}{M} \sum_{i=1}^{M} g_\theta^i(\mathbf{z}_e) \log g_\theta^i(\mathbf{z}_e), \tag{9}$$

$$\mathcal{L}_{\text{qua}} = \frac{1}{M} \sum_{i=1}^{M} g_\theta^i(\mathbf{z}_e) \cdot sg(\|\mathbf{z}_e - \underset{\mathcal{C}^i}{\text{Quant}}(\mathbf{z}_e)\|_2^2 - \bar{e}), \tag{10}$$

where $\bar{g}_\theta(\mathbf{z}_e)$ denotes the average probability of the current codebook group across samples in the batch, and $\bar{e} = \frac{1}{M} \sum_{i=1}^{M} \|\mathbf{z}_e - \text{Quant}_{\mathcal{C}^i}(\mathbf{z}_e)\|_2^2$. $\mathcal{L}_{\text{ent}}$ maximizes the entropy of selection distribution to enforce balanced utilization across all codebooks, preventing preferential collapse to dominant codebooks. This promotes full parameter space exploration during training. $\mathcal{L}_{\text{dec}}$ reduces prediction ambiguity by concentrating probability mass on the optimal codebook index. $\mathcal{L}_{\text{qua}}$ guides the router towards codebooks producing below-average reconstruction errors. The composite loss function becomes:

$$\mathcal{L}_{\text{router}} = \mathcal{L}_{\text{qua}} + \lambda_1 \mathcal{L}_{\text{ent}} + \lambda_2 \mathcal{L}_{\text{dec}}. \tag{11}$$

Although the learnable router $G_\theta$ provides adaptive codebook selection during training, its stochastic nature cannot guarantee persistent global optimality in routing decisions. This limitation stems from the exploration-exploitation dilemma inherent in entropy-regularized optimization. Consequently, we only employ $G_\theta$ for training. For inference, we only employ $G_{\text{naive}}$ to ensure quantization fidelity through guaranteed minimal-error codebook assignment.

## 3.4 Token-Specific Codebook Quantization

Previous vector quantization methods employ a global codebook $\mathcal{C}_{global} \in \mathbb{R}^{K \times d}$ where all tokens share the same quantization space. However, within individual facial images, local regions (e.g., ocular vs. nasal areas) exhibit significant feature-space divergence due to domain-specific texture and geometric variations.Therefore, individual tokens cannot effectively span the entire feature space, leading to incomplete utilization of the full codebook. Besides, token features may partially overlap, perfect alignment rarely occurs. As shown in Figure 2(c), we propose decomposing the global codebook into token-specific sub-codebooks:

$$\mathcal{C}_{\text{tsc}} = [\mathcal{C}_1 \oplus \mathcal{C}_2 \oplus \cdots \oplus \mathcal{C}_T] \in \mathbb{R}^{T \times K \times d}, \tag{12}$$

where $T$ denotes the number of tokens, each sub-codebook $\mathcal{C}_t \in \mathbb{R}^{K \times d}$ independently learns the distribution of the $t$-th token. The token-specific codebook quantization becomes:

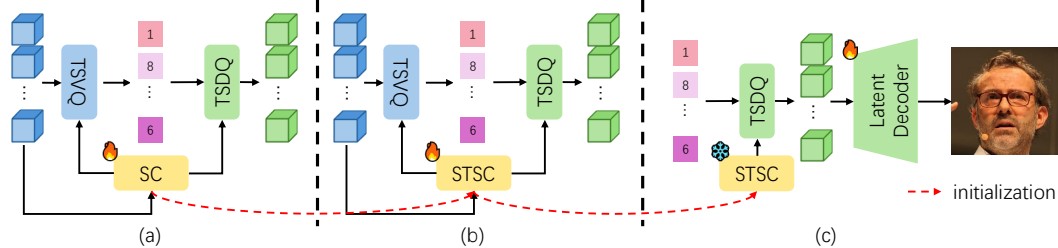

Figure 3: Illustration of the training strategy. (a) Training stage 1: Optimization of the switchable token-shared codebook. (b) Training stage 2: Optimization of switchable token-specific codebooks, initialized from the Stage 1 token-shared codebook. (c) Training stage 3: Exclusive latent decoder optimization with frozen switchable token-specific codebooks. Sample face images are from the FFHQ dataset (44).

$$\mathbf{z}_q^t = \underset{\mathcal{C}_t}{\mathrm{Quant}}(\mathbf{z}_e^t) = \underset{\mathbf{e} \in \mathcal{C}_t}{\arg\min} \|\mathbf{z}_e^t - \mathbf{e}\|_2^2. \tag{13}$$

Despite increased total codebook size ($T \times K$ vs $K$), per-token bit-width remains $b = \lceil \log_2 K \rceil$ identical to previous methods. Besides, by dedicating individualized sub-codebooks to model token-specific feature distributions, our method achieves higher sampling density within each token's characteristic subspace, which directly translates to improved reconstruction fidelity.

## 3.5 Training Strategy

Building upon our foundational innovations in Switchable Codebook and Token-Specific Codebook, we synergistically combine these components to formulate STSCQ — a novel architecture that leverages multiple token-specific codebooks to simultaneously reduce per-token index bit-width and mitigate quantization errors. To maximize synergistic advantages, as shown in Figure 3, we introduce a three-stage progressive training paradigm.

Firstly, we initialize our model with a pre-trained latent space model and implement Switchable Codebook Quantization — replacing the original single codebook with multiple learnable codebooks. After routing, all tokens within an image share a unified codebook for quantization. During this stage, we freeze all parameters except codebooks and routing network, focusing optimization on:

$$\mathcal{L}_{\mathrm{Stage1}} = \|\mathbf{z}_e - \underset{\mathcal{C}^i}{\mathrm{Quant}}(\mathbf{z}_e)\|_2^2 + \mathcal{L}_{\mathrm{router}}, \text{where } i = G_\theta(\mathbf{z}_e). \tag{14}$$

The first stage crucially establishes codebook diversity and routing policy initialization for subsequent token-specific adaptation. Building upon this foundation, the second stage implements progressive codebook refinement by leveraging the pre-trained codebooks as initialization vectors for token-specific codebooks. Following the same parameter freezing protocol as Stage 1, we exclusively update token-specific codebooks and routing network. The training objective during codebook refinement is:

$$\mathcal{L}_{\mathrm{Stage2}} = \sum_{j=1}^{T} \|\mathbf{z}_e^j - \underset{\mathcal{C}_j^i}{\mathrm{Quant}}(\mathbf{z}_e^j)\|_2^2 + \mathcal{L}_{\mathrm{router}}, \text{where } i = G_\theta(\mathbf{z}_e). \tag{15}$$

As the codebook's feature space evolves through training iterations, the pre-trained decoder becomes suboptimally aligned with the updated codebook representations. To maintain precise latent-to-pixel space mapping, we perform decoder fine-tuning on the original training dataset, ensuring accurate image reconstruction from the transformed latent features. Furthermore, given the critical requirement for high-fidelity preservation of facial attributes in compression systems, we integrate an identity conservation mechanism during decoder refinement. Specifically, we leverage the widely-adopted ArcFace (47) loss to impose semantic consistency between original and reconstructed faces. The decoder can be supervised by the image-level loss:

$$\mathcal{L}_{\text{Stage3}} = \|x - \hat{x}\|_2^2 + \lambda_p \mathcal{L}_{\text{per}} + \lambda_f \mathcal{L}_{\text{face}}, \tag{16}$$

where $\mathcal{L}_{\text{per}}$ denotes perception loss, $\mathcal{L}_{\text{face}}$ denotes face loss.

## 4 Experiments

### 4.1 Setups

**Dataset.** We train our models on the CASIA-WebFace dataset (48), a large-scale face image dataset widely used in the field of face recognition research, which contains approximately 500K images of 10,575 individuals, collected from the Internet. In the test stage, we evaluate the reconstruction quality of our method on five face recognition datasets: LFW (2), CFP-FP (49), AgeDB (50), CPLFW (51), CALFW (52). They all have 6-7K pairs of images that are used to determine whether they belong to the same person. Note that all images for training and evaluating are resized to $256 \times 256$, and data augmentation strategies such as random cropping and random flipping are applied during training.

**Training Details.** We adopt our Switchable Token-Specific Codebook Quantization on both CNN-based and ViT-based VQ-tokenizers. In our training pipeline, the encoder remains fixed throughout the process. During stage 1 and stage 2, only the codebook is learnable, with its initial size set to 4096. In stage 3, only the decoder is trained to adapt to the quantized representations produced by the updated codebook. For the training dataset CASIA-WebFace, we train 100K steps for stage 1, 400K steps for stage 2, and 100K steps for stage 3. Our models are optimized by AdamW with the initial learning rate of $1e - 4$. Our methods are implemented on eight NVIDIA V100 GPUs with nearly 2 days for training.

**Evaluation Metrics.** We evaluate the level of image compression using bits per pixel (bpp), and assess the impact of compression on facial images using a pre-trained face recognition model. Specifically, we compute the Mean Accuracy (MeanAcc) and the Identity Similarity (IDS). MeanAcc refers to the average recognition accuracy across five face recognition benchmark datasets after compression and reconstruction, while IDS measures the cosine similarity between the features of the original and reconstructed images.

### 4.2 Main Results

We evaluate our proposed method on two representative baselines, TiTok (7) and VQGAN (36). For TiTok, we conduct experiments under two different scales, where each image is represented by either 128 or 32 discrete indices. For VQGAN, we follow the experimental setup of MASKGIT (53). We conduct comparisons with a variety of state-of-the-art methods, encompassing both traditional compression algorithms (e.g., JPEG2000 (8)) and codebook-based learning approaches (e.g., MaskGIT and TiTok), as shown in Table 1. In comparison with other methods, our proposed approach preserves outstanding recognition effectiveness for compressed facial images, maintaining a recognition accuracy of around 70% even at extremely low bit rates (bpp < 0.01). Specifically, compared with traditional compression algorithms, our approach achieves higher recognition accuracy and IDS at approximately half the bit rate. However, at a compression rate of 0.01 bpp, JPEG2000 exhibits considerable limitations, as its capacity to retain essential facial details for reliable recognition is substantially reduced. For codebook learning-based methods, our approach also demonstrates outstanding performance. By learning token-specific codebooks for each token, we significantly enhance the representational capacity of the latent space. For example, on MaskGit-VQGAN, our method achieves a recognition accuracy of 93.51% and the IDS of 0.6659, while on TiTok-s, we obtain the accuracy of 91.66% and the IDS of 0.6120. These results represent a substantial improvement over the baseline methods.

### 4.3 Ablation Studies

**Generalization Ability of Our Method.** With the help of proposed Switchable Token-Specific Codebook Quantization, we can flexibly adjust the compression rate by reducing the size of codebooks while maximizing the retention of its latent space encoding capacity. To show generalizable design of our methods, we conduct generalization experiments on three baseline models, each evaluated

Table 1: Quantitative comparison with state-of-the-art methods.

| Method | Model Type | # Tokens | MeanAcc(%) | IDS | bpp |
|---|---|---|---|---|---|
| JPEG 2000 (8) | / | / | 56.98 | 0.0312 | 0.0100 |
| JPEG 2000 (8) | / | / | 85.64 | 0.3551 | 0.0500 |
| CodeFormer (41) | 2D | 256 | 89.99 | 0.6210 | 0.0390 |
| MaskGit-VQGAN (53) | 2D | 256 | 90.70 | 0.6314 | 0.0469 |
| TiTok-S (7) | 1D | 128 | 87.56 | 0.5764 | 0.0234 |
| TiTok-L (7) | 1D | 32 | 65.07 | 0.1812 | 0.0059 |
| Ours(MaskGit-VQGAN) | 2D | 256 | 93.51 (+2.81↑) | 0.6659 | 0.0469 |
| Ours(TiTok-S) | 1D | 128 | 91.66 (+4.10↑) | 0.6120 | 0.0234 |
| Ours(TiTok-L) | 1D | 32 | 73.13 (+8.06↑) | 0.2583 | 0.0059 |

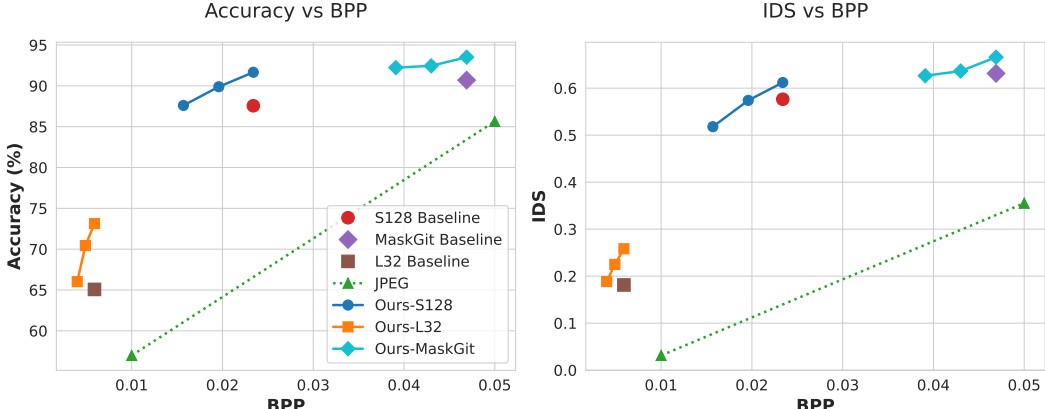

Figure 4: Comparisons of different baselines and our methods.

under three different bpp configurations. The results are shown in Table 2 and Figure 4. Our method demonstrates superior rate-distortion performance compared to conventional baselines across both 1D and 2D latent-space modeling frameworks. For 1D modeling (TiTok-S), we reduce the bitrate by 32.9% (from $0.0234 \to 0.0157$ bpp) while maintaining comparable recognition accuracy ($87.56\% \to 87.60\%$). In 2D latent-space architectures (MaskGit-VQGAN), our approach achieves a 16.6% bitrate reduction ($0.0469 \to 0.0391$ bpp) coupled with a 1.53% absolute accuracy improvement ($90.70\% \to 92.23\%$), validating that codebook specialization simultaneously enhances compression efficiency and feature representation fidelity.

To better evaluate the efficiency of our method, we evaluate the inference latency and storage overhead introduced by our method, as shown in Table 2. The expansion of the overall codebook size does indeed incur additional storage costs, along with a slight increase in inference latency. However, we explore a routing-based inference: only the codebook group selected by the router needs to be loaded, rather than searching with the minimum error across all codebooks. With this improvement, both inference latency and storage overhead are substantially alleviated, while the performance remains competitive with the baselines.

**Effectiveness of Switchable Codebook Quantization.** We conduct ablation studies about our proposed method with the codebook size of 1024 and show results in Table 3, where Idx0 indicates the original single codebook without any modifications, and NN means the nearest-neighbor search. The results for Idx0 and Idx1 indicate that employing a routing mechanism to select among multiple codebooks can further enhance the representational capacity of the codebooks under a fixed bpp, as evidenced by an improvement in recognition accuracy from 88.11% to 88.24%.

**Effectiveness of Codebook Routing and Token-Specific Codebook.** As shown in Table 3, the results for Idx2 and Idx4 suggest that the presence of multiple codebooks, together with the routing mechanism, can further alleviate the learning difficulty in the feature quantization process and improve model performance. In addition, the comparison between Idx1 and Idx4 demonstrates that

Table 2: Switchable Token-Specific Codebook Quantization on different baselines. $\#Tks$ and $\#Cbs$ indicates the number of tokens and codebooks respectively. $N_{Cb}$ refers to the size of each codebook. 'SP' means selection policy, where 'NN' means Nearest-Neighbor search, 'CR' means proposed Codebook Routing search. $t_{inf}$ indicates the inference time per image tested on a V100.

| Backbone | $\#Tks$ | $\#Cbs$ | $N_{Cb}$ | bpp | SP | MeanAcc(%) | IDS | $t_{inf}$ (s) | Storage (MB) |
|---|---|---|---|---|---|---|---|---|---|
| MaskGit-VQGAN (53) | 256 | 1 | 4096 | 0.0469 | NN | 93.51 | 0.6659 | 0.1554 | 1122.70 |
| | 256 | 16 | 1024 | 0.0391 | NN | 92.23 | 0.6264 | 0.1771 | 4194.70 |
| | 256 | 16 | 1024 | 0.0391 | CR | 92.22 | 0.6253 | 0.1544 | 354.70 |
| TiTok-S (7) | 128 | 1 | 4096 | 0.0234 | NN | 91.66 | 0.6120 | 0.1437 | 122.68 |
| | 128 | 256 | 256 | 0.0157 | NN | 87.60 | 0.5180 | 0.1496 | 482.71 |
| | 128 | 256 | 256 | 0.0157 | CR | 87.54 | 0.5125 | 0.1450 | 100.21 |
| TiTok-L (7) | 32 | 1 | 4096 | 0.0059 | NN | 73.13 | 0.2583 | 0.1744 | 1163.47 |
| | 32 | 256 | 256 | 0.0040 | NN | 66.02 | 0.1885 | 0.1750 | 1253.50 |
| | 32 | 256 | 256 | 0.0040 | CR | 65.65 | 0.1864 | 0.1741 | 1157.87 |

Table 3: Ablation studies.

| Idx | Tok-shared | Tok-specific | NN | CR | MeanAcc(%) | IDS | bpp |
|---|---|---|---|---|---|---|---|
| 0 | - | - | - | - | 88.11 | 0.5361 | 0.0195 |
| 1 | ✓ | - | - | ✓ | 88.24 | 0.5412 | 0.0196 |
| 2 | - | ✓ | ✓ | - | 89.28 | 0.5701 | 0.0196 |
| 3 | - | ✓ | - | ✓ | 89.89 | 0.5740 | 0.0196 |

learning a specific codebook for each token enhances the representational capacity of each token in the latent space, as evidenced by improvements in recognition accuracy. Furthermore, token-specific codebook quantization is able to solve the uneven distribution of codebook utilization due to the original strategy of using a global-shared codebook across all tokens. As shown in Table 4, the proposed approach enables more effective utilization of the codebook, with an average increase of approximately 20% in per-token codebook usage, thereby reducing quantization errors caused by codebook utilization imbalance.

Table 4: Codebook utilization rates (%) per token on LFW dataset.

| Method | bpp | Min | Max | Mean | STD |
|---|---|---|---|---|---|
| Global-shared | 0.0234 | 3.49 | 78.12 | 54.17 | 14.71 |
| Ours | 0.0234 | 17.90 | 83.89 | 74.02 | 9.14 |

## 5 Conclusion and Limitation

In this paper, we propose a switchable token-specific codebook quantization mechanism. Specifically, we design a codebook routing algorithm that assigns each image to its own small codebook, and further allocate an independent codebook to each token within the image. Our approach supports flexible bpp (bits-per-pixel) settings and enables the codebook to better exploit its representational capacity under the same bpp configuration. We validate the effectiveness of our method on face recognition tasks, demonstrating that facial images can maintain competitive recognition accuracy even when compressed to extremely low bpp.

However, our method still has certain limitations. Since our approach focuses on flexible configuration of codebook size without introducing special designs for the encoder or decoder, its performance is highly dependent on the underlying autoencoder used as the baseline. We leave these extensions for future work.

**Acknowledgements.** This work is supported by the National Natural Science Foundation of China No. 62476090, 62302167, U23A20343, 62222602, 62176092, 72192821; Shanghai Sailing Program 23YF1410500; Young Elite Scientists Sponsorship Program by CAST YESS20240780; Natural Science Foundation of Chongqing CSTB2023NSCQJQX0007, CSTB2023NSCQ-MSX0137; CCF-Tencent RAGR20240122; the Open Research Fund of Key Laboratory of Advanced Theory and Application in Statistics and Data Science-MOE, ECNU.

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

# A  Technical Appendices and Supplementary Material

## A.1  Image-level and Token-level Analysis

Due to the unique characteristics of facial images, images sharing the same attributes (such as ethnicity, gender, etc.) often exhibit many common features at the image level. At the token level, tokens corresponding to different facial regions also tend to have distinct feature representations. To address the attribute distribution characteristics at the image level, we design multiple groups of codebooks with a routing mechanism to capture both the differences and commonalities among image attributes. As illustrated in Figure 5a, we visualize the activation patterns of 16 codebook groups with respect to the ethnicity attribute. It can be observed that, for African faces, the 5th and 14th codebook groups are frequently activated, whereas for Asian faces, the 9th and 3rd codebook groups are more likely to be activated. This observation supports our hypothesis that designing separate codebooks for different attributes is beneficial.

At the token level, we conducted a statistical analysis comparing two approaches: sharing a single codebook among all tokens versus learning a separate codebook for each token, as shown in Figure 5b. The results indicate that when all tokens share a single codebook, tokens at different positions are unable to fully exploit the representational capacity of the codebook's latent space. This is reflected in the low utilization rate and large standard deviation for individual tokens. In contrast, with our proposed token-specific codebook approach, the utilization of the codebook by each token is significantly improved, with consistently high utilization rates across all tokens.

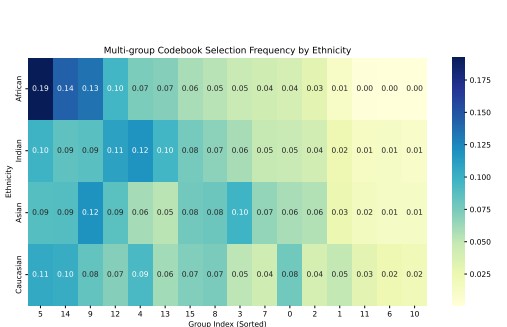

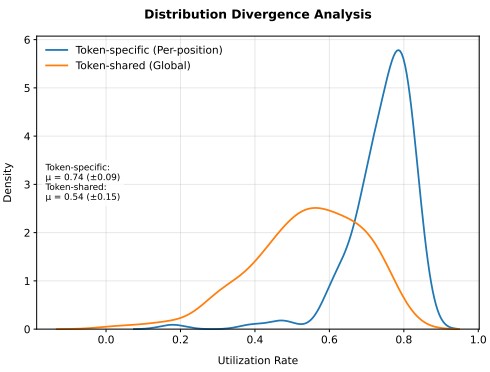

(a) Visualization analysis of codebook routing on different ethnics.

(b) Visualization analysis of codebook utilization of different tokens.

Figure 5: Visualization analysis from image-level and token-level.

## A.2  Alleviating the Trade-off between Compression and Recognition Accuracy

Directly reducing the codebook size to lower the bpp often leads to significant performance degradation. In contrast, our method effectively mitigates the loss in recognition performance associated with decreasing bpp. As shown in Figure 6, we conducted experiments on the TiTok-s128 baseline under three different bpp settings. The results demonstrate that our approach consistently improves recognition performance across all bpp levels. Moreover, as bpp decreases, our method better preserves recognition accuracy. For example, at bpp = 0.0235, our method achieves improvements of 0.97% in recognition accuracy and 0.019 in IDS compared to the baseline. At an even lower bpp of 0.0157, the improvements increase to 1.32% in accuracy and 0.043 in IDS. These results confirm the robustness of our method to changes in bpp.

## A.3  Impact of the Number of Routing Codebooks

Our method employs multiple small codebooks and utilizes a routing mechanism to assign each image to its respective codebook. This approach enables a reduction in codebook size while maximally preserving the representational capacity of the codebooks' latent space. To investigate the impact of the number of codebooks on final recognition performance, we conducted further experiments. As

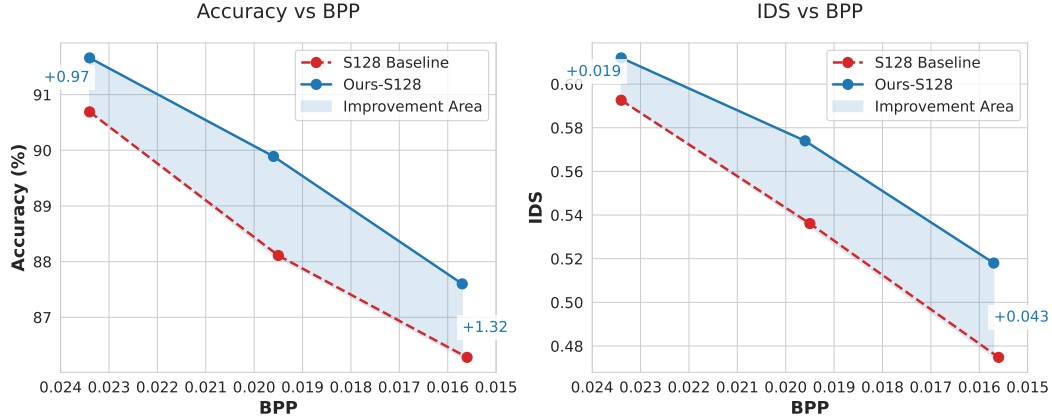

Figure 6: Comparison of recognition performance degradation with decreasing bpp between our method and the baseline.

shown in Figure 7, with the codebook size fixed at 1024, increasing the number of codebooks from 1 to 16 leads to a noticeable improvement in recognition accuracy. This suggests that increasing the number of codebooks can indeed mitigate the performance degradation caused by directly reducing codebook size. However, we also observed that when the number of codebooks is further increased to 128, recognition performance begins to decline. We speculate that this is due to the increased learning difficulty associated with a large number of codebooks. Fully leveraging the vast representational capacity of such a large latent space is beyond the scope of this work.

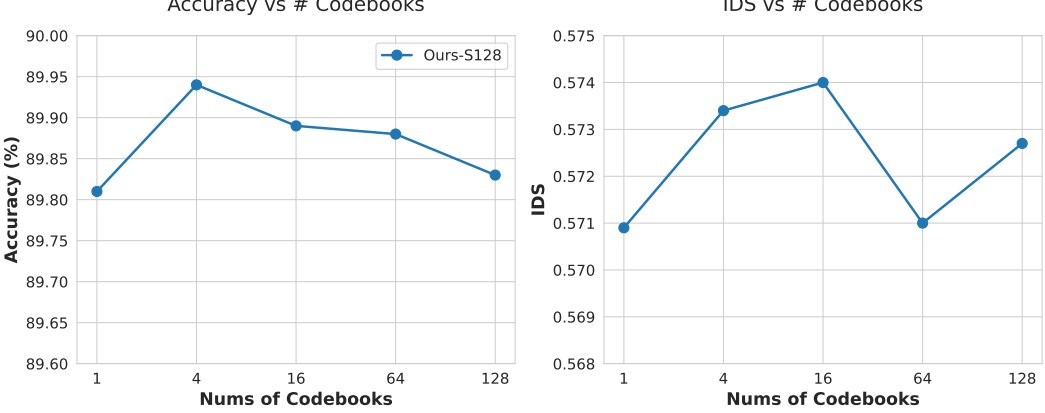

Figure 7: Effect of routing codebook quantity on recognition accuracy.

