# OpenReview forum: "Switchable Token-Specific Codebook Quantization For Face Image Compression"
_NeurIPS.cc/2025/Conference — NeurIPS 2025 poster_

### Official Review · Reviewer_twLH · 2025-06-03

**Clarity:** 3
**Significance:** 3
**Originality:** 2
**Rating:** 5
**Confidence:** 4

**Summary:**

This paper proposes a novel method for compressing facial images using a Switchable Token-Specific Codebook Quantization (STSCQ) framework. Traditional VQ-based image compression methods degrade under ultra-low bitrate (bpp) due to reliance on a single global codebook. To address this, the authors introduce a two-level codebook structure: (i) image-level routing to select a specialized codebook and (ii) token-level codebook decomposition to assign each token its own quantizer. The architecture is trained in a three-stage progressive manner, incorporating ArcFace loss for identity preservation. Extensive experiments across five face recognition benchmarks show improved rate-distortion trade-offs and recognition accuracy over prior methods like TiTok and MaskGIT. The method demonstrates superior compression efficiency and recognition fidelity, particularly at low bpp (e.g., 66.02% accuracy at 0.004 bpp).

**Questions:**

1. Can you rerun the full evaluation over ≥ 3 random seeds and report mean ± standard deviation (or 95 % confidence intervals) for all metrics in Tables 1 and 2?
2. What are the training-time and inference-time costs, FLOPs, and latency, relative to TiTok-s and MaskGIT?

**Ethical Concerns:**

["NO or VERY MINOR ethics concerns only"]

**Final Justification:**

The authors’ rebuttal offers a generally adequate and structured response to the raised concerns, addressing each key issue with new experimental evidence and clarifications. For statistical robustness, they reran experiments over three random seeds and reported mean ± standard deviation for the TiTok-S baseline. While this partially satisfies the request, the omission of multi-seed evaluations for other backbones limits completeness, though they reasonably justify this with resource constraints.

The concern regarding computational overhead is fully addressed: detailed tables report inference time, GPU memory, storage, and FLOPs across configurations, comparing their method with TiTok and MaskGIT and demonstrating significant improvements, particularly under the Codebook Routing strategy.

On demographic fairness, the authors provide per-ethnicity accuracy evaluations using the RFW dataset and argue that the high number of codebooks dilutes any strong alignment with specific groups. While no direct mitigation is implemented, the authors propose future work involving group-specific decoder fine-tuning, which reasonably addresses the concern.

Ethical risks around ultra-low-bpp face compression are now acknowledged, with promised countermeasures such as watermarking and usage guidelines to be included in the final version. Although the authors do not explicitly expand on the originality concern, they reiterate the architectural innovations of their token-specific quantization and routing strategy.

Overall, the rebuttal resolves the primary issues of evaluation rigor, computational efficiency, and ethical responsibility. The rebuttal justifies an upgrade to an accept.

**Limitations:**

No. The paper lists technical limitations (dependency on the underlying auto-encoder) and briefly claims positive societal value, but it omits two key areas:

1. Supplementary Fig. 2 shows that routing groups correlate with ethnicity; however, the authors provide no demographic parity analysis or mitigation strategy. A brief experiment on RFW or a similar dataset, along with a discussion of safeguards (e.g., bias-aware training or post-compression calibration), is necessary.

2. Ultra-low-bpp face compression can lower the cost of large-scale surveillance. The paper should acknowledge this and outline responsible-release measures (e.g., gating access to pretrained weights, adding watermarking, or providing usage guidelines).

**Quality:**

3

**Strengths And Weaknesses:**

The paper offers a compelling, though not flawless, contribution. Its chief strength lies in the originality of a two-level quantization design that marries switchable image-level routing with token-specific sub-codebooks, an architectural twist that demonstrably boosts face-recognition accuracy at ultra-low bitrates and thus promises real utility for storage-constrained biometric pipelines.

Methodologically, the authors conduct sensible ablations, test the idea on both 1-D and 2-D latent tokenizers, and document a clear three-stage training schedule. Well-designed figures also aid reader comprehension.

However, important weaknesses remain: all quantitative results lack error bars or confidence intervals, so the statistical robustness of the reported gains is uncertain. Furthermore, the novelty of image-level routing is somewhat incremental given prior work on mixture-of-experts and product-quantization, and the paper omits a discussion of computational overhead, which is crucial for edge deployment.

If the authors add statistical significance reporting and quantify the computational cost, the work would merit acceptance; without these corrections, the contribution falls just short of the bar for a premier venue.

---

> ### Author Rebuttal · Authors · 2025-07-31
>
> **Answer 1:**
>
> We re-evaluated our method using three different random seeds, and the results are presented in the table below. Due to time and resource constraints, we are only able to report results based on the TiTok-S baseline. As shown in the table, the impact of random seed variation is not significant, which can be attributed to the fact that the core of our method lies in training the codebook, while the encoder and decoder are initialized with pre-trained weights. We plan to update the re-evaluation results for other baselines in future versions.
>
> **Table: Switchable Token-Specific Codebook Quantization on different baselines.**
>
> | Backbone  | # Tokens | # Codebooks | Codebook Size | MeanAcc(%)         | IDS                | Storage(bits) | bpp    |
> |-----------|----------|-------------|---------------|--------------------|--------------------|--------------|--------|
> | TiTok-s   | 128      | 1           | 4096          | 91.65±0.02         | 0.6130±0.0020      | 1536         | 0.0234 |
> | TiTok-s   | 128      | 16          | 1024          | 89.75±0.14         | 0.5681±0.0059      | 1284         | 0.0196 |
> | TiTok-s   | 128      | 256         | 256           | 87.63±0.03         | 0.5179±0.0002      | 1032         | 0.0157 |
>
> **Answer 2:**
>
> We have reported the FLOPs of the model, the required storage overhead and the inference time per image on a V100 GPU in the table below. As shown, our method does introduce additional storage and inference latency, and this effect becomes more pronounced as the number of codebook groups and the size of the codebook latent space increase. We attribute this to the need to maintain all codebook groups during inference when selecting the nearest codebook based on the minimum error criterion. To address this, we experimented with directly using the routing results during inference, so that the decoder only needs to access the codebook corresponding to the routing index. The benefits of this approach are clearly reflected in the table: for example, with the MaskGIT backbone, storage overhead is reduced by approximately 60\% and inference latency by about 40\%, while remaining the performance of MeanAcc and IDS.
>
> **Table: Inference latency and memory overhead statistics. For ours method, 'NN' and 'CR' refer to Nearest-Neighbor search and Codebook Routing search respectively.**
>
> |Backbone|bpp|Token-specific|Codebooks|Infer Time (s / img)|GPU Mem (MB)|Storage (MB)|FLOPs|
> |-|-|-|-|-|-|-|-|
> |Maskgit-VQGAN|0.0469|-|1|0.1495|474.14|94.69|158.14 G|
> |Ours (NN)|0.0469|✓|1|0.1554|3283.39|1122.70|158.14 G|
> |Ours (NN)|0.0391|✓|16|0.1771|12508.17|4194.70|158.14 G|
> |Ours (CR)|0.0391|✓|16|0.1544|5116.15|354.70|158.14 G|
> |TiTok-S|0.0234|-|1|0.1433|586.03|98.68|113.57 G|
> |Ours (NN)|0.0234|✓|1|0.1437|609.84|122.68|113.57 G|
> |Ours (NN)|0.0157|✓|256|0.1496|1894.54|482.71|113.57 G|
> |Ours (CR)|0.0157|✓|256|0.1450|1157.62|100.21|113.57 G|
> |TiTok-L|0.0059|-|1|0.1728|2709.24|1157.61|269.62 G|
> |Ours (NN)|0.0059|✓|1|0.1744|2716.80|1163.47|269.62 G|
> |Ours (NN)|0.0040|✓|256|0.1750|2865.43|1253.50|269.62 G|
> |Ours (CR)|0.0040|✓|256|0.1741|2804.02|1157.87|269.62 G|
>
> **Table: Quantitative comparison between different codebook selection policies during inference.**
>
> |Backbone|Codebook Selection|bpp|MeanAcc(%)|IDS|
> |-|-|-|-|-|
> |Maskgit-VQGAN|NN|0.0391|92.23|0.6264|
> |Maskgit-VQGAN|CR|0.0391|92.22|0.6253|
> |TiTok-S|NN|0.0157|87.60|0.5180|
> |TiTok-S|CR|0.0157|87.54|0.5125|
> |TiTok-L|NN|0.0040|66.02|0.1885|
> |TiTok-L|CR|0.0040|65.65|0.1864|
>
> **Answer for limitations 1:**
>
> We conducted experiments on the four ethnic groups in the RFW dataset, and the results are shown in the table below. Our findings indicate that the introduction of multiple codebooks does not lead to significant changes in ethnic bias. Although our supplementary materials reveal that the codebook groupings are related to ethnicity, the number of codebook groups is much larger than the number of ethnic categories, so each group is not strongly associated with any single ethnicity. We suggest that future work could explore fine-tuning the decoder for each ethnic group in our stage 3 to better balance recognition performance across different groups. We will include a discussion of these limitations in future versions.
>
> **Table: Quantitative comparison between different codebook selection policies during inference.**
>
> | Backbone | bpp    | # Codebooks | African | Asian  | Caucasian | Indian | MeanAcc(%)        | IDS    |
> |----------|--------|-------------|---------|--------|-----------|--------|-------------------|--------|
> | MaskGIT  | 0.0469 | 1           | 83.30   | 83.12  | 90.22     | 85.13  | 85.44±2.87        | 0.6932 |
> | MaskGIT  | 0.0391 | 16          | 80.35   | 79.27  | 87.72     | 84.70  | 83.01±3.39        | 0.6592 |
> | TiTok-S  | 0.0234 | 1           | 80.13   | 80.00  | 87.88     | 87.65  | 82.42±3.22        | 0.6428 |
> | TiTok-S  | 0.0157 | 256         | 71.97   | 76.52  | 79.25     | 75.78  | 75.88±2.60        | 0.5438 |
>
> **Answer for limitations 2:**
>
> We acknowledge that ultra-low-bpp face compression carries the risk of facilitating large-scale surveillance and potential misuse. To mitigate such concerns, we will provide clear usage guidelines and ethical instructions upon the open-sourcing of our project, and incorporate watermarking into the compressed images to help prevent improper use. We will include a statement addressing these issues in the limitations section of a future version.

---

> > ### Comment · Reviewer_twLH · 2025-08-05
> >
> > Overall, the rebuttal resolves the primary issues of evaluation rigor, computational efficiency, and ethical responsibility. The rebuttal justifies an upgrade to an accept.

---

### Official Review · Reviewer_GXvd · 2025-06-27

**Clarity:** 2
**Significance:** 2
**Originality:** 3
**Rating:** 3
**Confidence:** 4

**Summary:**

This paper proposes a switchable token-specific codebook quantization method which dynamically specifies a codebook for each image using router and uses specified sub-codebooks for each token in the codebook. The hierarchically dynamic structure significantly reduces the required bpp while achieving high accuracy on face recognition datasets.

**Questions:**

1.From the title of the paper, it appears to be a paper in the field of facial image encoding. But from the content of the article, it seems more like a downstream task in the field of facial image encoding. I recommend supplementing the experimental results according to the weakness on experiment mentioned in the previous text.
2.As the point mentioned in the weakness on method,  in Equation (8) and  in Equation (9) are essentially the same. The effectiveness of these two losses should be clearly discussed in the paper. Moreover, what are the consequences of using different routing strategies during training and inference due to the optimization dilemmas?
It would be more convincing if the authors could address the issues related to their methods

**Ethical Concerns:**

["NO or VERY MINOR ethics concerns only"]

**Final Justification:**

I maintain my original score. The image quality achieved by the proposed method does not appear to show significant improvement compared to the baseline. I understand that the scope of this work is focused on face recognition; however, I suggest revising the paper’s title to be more specific, as “face image compression” implies a substantially different concept in the compression domain. Given that the title explicitly claims “face image compression,” a thorough comparison with a broad range of existing methods in this area is expected. Currently, such comparisons are too limited—I have listed a few examples, but ideally, many more should be included. Moreover, the experiments primarily emphasize the low-bpp setting, without addressing whether the method can effectively compress high-resolution images. The papers cited mostly operate at very low bpp, and some are specifically designed for ID feature preservation, which can lead to particularly strong results in those conditions. As it stands, the compression application scenario targeted by this work appears very narrow. If this is positioned as a compression paper, from the perspective of the compression community, I do not consider it to fully meet that standard.

**Limitations:**

Yes

**Quality:**

3

**Strengths And Weaknesses:**

Paper Strengths:
1.The paper addresses an important challenge in the low bpp facial image compression by proposing a dynamic method for specifying codebooks.
2.The paper is generally easy to follow, and the mathematical formulations are logically consistent.
3.Quantitative experiments demonstrate the advantages of the proposed method over existing baselines.

Major Weaknesses:
Weakness on paper writing:

1.In Equation (12), the subscripts of the first two codebooks are both 1, which should be modified to 1 and 2.
2.In line 284-285, the original text in the paper is: “our approach achieves a 16.6% bitrate reduction (0.0469 → 0.0391 bpp) coupled with a 1.53% absolute accuracy improvement (90.70% → 92.23%).” The accuracy “90.70%” is inconsistent with the data “93.51%” provided in Table 3. Please check the validity of the data.

Weakness on method:
1.In order to achieve dynamic routing of the codebooks, the paper introduces a router to calculate the probability distribution of each codebook. However, the paper points out that the routing strategy is inconsistent during training and inference due to the exploration-exploitation dilemma inherent in entropy-regularized optimization, which is equivalent to the router only working during training. The paper lacks theoretical analysis of the dilemma. It would be beneficial to discuss the reason of the dilemma.
2.In Equation (8) and Equation (9), two losses designed for training routers are introduced, which are calculated based on the probability distribution output by the router. However, due to the lack of description of  in the text, based on the known information, these two losses are actually equivalent to the information entropy of selection distributions with different weights.

Weakness on experiments:
1.For evaluating compression performance, the paper only relies on Mean Accuracy and IDS, ignoring commonly used evaluation metrics in facial image compression, such as PSNR and LPIPS. It is recommended to complete experimental results to obtain a more comprehensive evaluation.
2.The paper mentions that five facial datasets were used for evaluation in the baseline model comparison experiment. However, the paper did not provide specific results on each dataset. At the same time, it is recommended to expand the number of baseline models, which helps to more accurately evaluate the effectiveness of the method
3.The paper should compare with some facial compression methods such as [1][2][3].

Reference:
[1] Qi Mao et al. Scalable Face Image Coding via StyleGAN Prior: Towards Compression for Human-Machine Collaborative Vision. TIP.2023
[2] Yuefeng Zhang et al. Machine Perception-Driven Facial Image Compression: A Layered Generative Approach. TCSVT.2024.
[3] Shuai Yang et al. Towards Coding for Human and Machine Vision:: Scalable Face Image Coding.TMM.

---

> ### Author Rebuttal · Authors · 2025-07-31
>
> **Answer for paper writing:**
>
> Thank you for pointing out the editorial error; we will correct it in future revisions. Specifically, regarding lines 284-285, our intention was to convey that our method is able to maintain—or even surpass—the baseline performance as the bpp decreases. For example, when bpp = 0.0469, the baseline achieves a MeanAcc of 90.70\%, whereas our method achieves a MeanAcc of 92.23\% at a lower bpp of 0.0391. We hope this clarification addresses your concerns.
>
> **Answer for method 1:**
>
> Selecting the most appropriate codebook for the model poses a significant challenge. During training, we experimented with directly using a nearest neighbor strategy for codebook selection. However, we observed that this approach resulted in a notably low codebook utilization rate of only 19.89\%, indicating that the multiple codebooks were not being fully exploited. We attribute this to the nearest neighbor strategy’s tendency to favor codebooks that have been more thoroughly optimized, leading to a collapse where multiple codebooks degenerate into a single codebook.
>
> **Table: Comparisons of different codebook selection strategies used while training.**
>
> | Method| Codebook Selection | # Tokens | # Codebooks | Codebook Size | MeanAcc(%) | IDS    | Codebook Mean Utilization(%) |
> |-|-|-|-|-|-|-|-|
> | Ours(TiTok-S) | NN| 128| 16| 1024| 89.28| 0.5701 | 19.89|
> | Ours(TiTok-S) | Routing| 128| 16| 1024| 89.89| 0.5740 | 89.34|
>
> To address this issue, we designed a routing strategy based on entropy-regularized optimization during training, which encourages the model to select a more diverse set of codebooks and mitigates codebook overfitting, thereby promoting more comprehensive codebook training. During inference, we adopt the nearest neighbor strategy based on the minimum loss principle, as the codebook with the smallest loss corresponds to the upper bound of our model’s performance. We also experimented with using the routing strategy during inference and found that its performance was comparable to that of the nearest neighbor strategy, while offering a more efficient inference process. Specifically, it significantly reduced both inference latency and storage overhead. For example, under the TiTok-S architecture, the inference latency per image using the routing strategy was 0.0103 seconds, while remaining the MeanAcc of 87.54\%, compared to 0.0154 seconds with the nearest neighbor strategy for the MeanAcc of 87.60\%.
>
> **Table: Quantitative comparison between different codebook selection policies during inference.**
>
> |Backbone|Codebook Selection|bpp|MeanAcc(%)|IDS|
> |-|-|-|-|-|
> |Maskgit-VQGAN|NN|0.0391|92.23|0.6264|
> |Maskgit-VQGAN|CR|0.0391|92.22|0.6253|
> |TiTok-S|NN|0.0157|87.60|0.5180|
> |TiTok-S|CR|0.0157|87.54|0.5125|
> |TiTok-L|NN|0.0040|66.02|0.1885|
> |TiTok-L|CR|0.0040|65.65|0.1864|
>
> **Answer for method 2:**
>
> We apologize for the oversight—there is an error in Equation 8 in the manuscript. The correct form is $\mathcal{L}\_{\text{ent}} = \sum\_{i=1}^M \bar{g\_{\theta}}(\mathbf{z}\_e) \log \bar{g\_{\theta}}(\mathbf{z}\_e)$, where $\bar{g\_{\theta}}(\mathbf{z}\_e)$ denotes the average probability of the current class across samples in the batch. Equation 8 is designed to encourage the router to select diverse codebook groups for different samples within the same batch, thereby fully exploring and utilizing the expanded codebook latent space. In contrast, Equation 9 measures the entropy of the predicted distribution for each individual sample, aiming to suppress excessive uncertainty in single-sample predictions. Together with Equation 10, these three objectives not only promote comprehensive utilization of the codebook space, but also effectively balance inter-sample diversity with the certainty of individual predictions.
>
> **Answer for experiment 1:**
>
> You are absolutely right that including image quality metrics is essential for a comprehensive evaluation. In response, we have additionally assessed both 1D and 2D baselines, using three widely adopted metrics: SSIM, PSNR, and LPIPS. The results are summarized in the table below, where IBQ refers to the state-of-the-art 2D baseline published at ICCV 2025. As shown, our method achieves significant improvements in image quality, further demonstrating its effectiveness. Moreover, we observe that the LPIPS metric fluctuates across different baselines. We hypothesize that this is because the LPIPS score is computed using a network pre-trained on ImageNet to assess perceptual similarity between images, while there are substantial differences between facial data and ImageNet data. Therefore, we prefer to use MeanAcc and IDS as more reliable indicators of machine perception for facial compression tasks. We will incorporate these latest results in the revised version of our paper.
>
> **Table: Quantitative comparison about image quality with baseline methods.**
> | Method| Model Type | # Tokens | bpp    | SSIM ↑  | PSNR ↑ | LPIPS ↓ |
> |-|-|-|-|-|-|-|
> | IBQ            | 2D         | 256      | 0.0508 | 0.8520  | 29.79  | 0.1880  |
> | MaskGIT        | 2D         | 256      | 0.0469 | 0.7908  | 27.22  | 0.2204  |
> | TiTok-S        | 1D         | 128      | 0.0234 | 0.7799  | 27.21  | 0.2323  |
> | TiTok-L        | 1D         | 32       | 0.0059 | 0.6868  | 23.25  | 0.2959  |
> | Ours(IBQ)      | 2D         | 256      | 0.0508 | 0.8600  | 30.21  | 0.1765  |
> | Ours(MaskGIT)  | 2D         | 256      | 0.0469 | 0.7802  | 27.27  | 0.2395  |
> | Ours(TiTok-S)  | 1D         | 128      | 0.0234 | 0.7823  | 27.39  | 0.2354  |
> | Ours(TiTok-L)  | 1D         | 32       | 0.0059 | 0.6931  | 23.33  | 0.2968  |
>
> **Answer for experiment 2:**
>
> We have provided the detailed recognition accuracy and IDS metrics on five face recognition datasets, as shown in the table below.
>
> **Table: Recognition accuracy(%) on five facial datasets.**
> | Method         | Model Type | bpp    | LFW   | CFP_FP | AgeDB | CPLFW | CALFW | Mean   |
> |----------------|------------|--------|-------|--------|-------|-------|-------|--------|
> | MaskGIT        | 2D         | 0.0469 | 98.33 | 89.11  | 88.03 | 88.33 | 89.67 | 90.70  |
> | TiTok-S        | 1D         | 0.0234 | 96.51 | 85.29  | 83.41 | 86.15 | 86.45 | 87.56  |
> | TiTok-L        | 1D         | 0.0059 | 77.84 | 63.07  | 57.29 | 62.19 | 64.97 | 65.07  |
> | Ours(MaskGIT)  | 2D         | 0.0469 | 99.46 | 92.40  | 91.09 | 91.82 | 92.79 | 93.51  |
> | Ours(TiTok-S)  | 1D         | 0.0234 | 98.68 | 90.21  | 88.77 | 89.12 | 91.52 | 91.66  |
> | Ours(TiTok-L)  | 1D         | 0.0059 | 87.34 | 67.07  | 65.74 | 71.82 | 73.66 | 73.13  |
>
> **Table: IDS on five facial datasets.**
>
> | Method         | Model Type | bpp    | LFW    | CFP_FP | AgeDB  | CPLFW  | CALFW  | Mean   |
> |----------------|------------|--------|--------|--------|--------|--------|--------|--------|
> | MaskGIT        | 2D         | 0.0469 | 0.6481 | 0.5989 | 0.6198 | 0.6643 | 0.6259 | 0.6314 |
> | TiTok-S        | 1D         | 0.0234 | 0.5883 | 0.5558 | 0.5570 | 0.6076 | 0.5733 | 0.5764 |
> | TiTok-L        | 1D         | 0.0059 | 0.1719 | 0.1682 | 0.1743 | 0.2215 | 0.1696 | 0.1812 |
> | Ours(MaskGIT)  | 2D         | 0.0469 | 0.6751 | 0.6428 | 0.6765 | 0.6654 | 0.6698 | 0.6659 |
> | Ours(TiTok-S)  | 1D         | 0.0234 | 0.6187 | 0.5967 | 0.6180 | 0.6072 | 0.6195 | 0.6120 |
> | Ours(TiTok-L)  | 1D         | 0.0059 | 0.2563 | 0.2381 | 0.2405 | 0.2963 | 0.2604 | 0.2583 |
>
> Regarding your suggestion to compare with face compression methods, we have carefully reviewed the three representative papers you mentioned. Unfortunately, none of these works have released their code, and their experimental settings for the face compression task differ from ours. Specifically, these studies focus on preserving image details in high-resolution data. For example, [1] trains on the high-resolution FFHQ dataset and evaluates image quality on CelebA-HQ, which is not directly comparable to our setting, where the goal is to compress images from face recognition datasets while preserving identity features.
>
> Nevertheless, we attempted to use screenshots of the visual results in Fig.6 from [2] and computed the IDS value between the original images and compressed images to assess the effectiveness in preserving facial identity. As shown in the table, our method demonstrates superior performance in identity preservation. In future versions, we will also include visual comparisons with these face compression methods.
>
> **Table: Comparison of IDS between other facial compression methods and ours.**
>
> | Method         | bpp    | IDS ↑   |
> |----------------|--------|---------|
> | MPDFIC[2]      | 0.246  | 0.4788  |
> | MPDFIC[2]      | 0.092  | 0.3571  |
> | MPDFIC[2]      | 0.045  | 0.1866  |
> | Ours(TiTok-S)  | 0.0234 | 0.6357  |
> | Ours(TiTok-S)  | 0.0157 | 0.5626  |

---

### Official Review · Reviewer_whZv · 2025-07-02

**Clarity:** 2
**Significance:** 2
**Originality:** 3
**Rating:** 4
**Confidence:** 3

**Summary:**

The authors propose a new vector quantization approach for low-bitrate image compression named Switchable Token-Specific Codebook Quantization (STSCQ). The approach addresses a key limitation of existing VQ-based compression models, where a globally shared codebook suffers at low bitrates due to reduced size. STSCQ introduces both image-level switchable codebooks (trained via routing) and token-specific sub-codebooks to better adapt to spatial and identity-specific variations in facial images. The method is evaluated on top of TiTok (Yu et al., 2024) and VQGAN (Esser et al., 2021), demonstrating improved performance on facial image reconstruction and recognition tasks at very low bpp.

**Questions:**

- In eq. 8, what is $p_i( \cdot )$? The entropy loss seems incorrectly formulated, should it be $\sum_i p_i \log p_i$?
- In eq. 12, the second $C_1$ seems a typo, should it be $C_2$? Also, does this imply that each of the $M$ image-level codebooks contains $T$ token-specific codebooks, for a total of $M \times T$ sub-codebooks, each being the same as $C_{global}$?
- While the motivation for token-specific codebooks is reasonable, would assigning a separate codebook to each token lead to redundancy? Have you measured codebook utilization for each $C^j_i$? Are they all being used effectively?
- For the main results (Table 1), how are the codebook size $K$ and the number of image-level codebooks $M$ selected?
- In token-specific quantization, is only nearest-neighbor search (eq. 6) used? Why not extend the routing mechanism to this level as well to improve diversity?
- The authors mention that the router is used only during training due to exploration-exploitation tradeoffs. Have you compared the posterior codebook utilization between learned routing and nearest-neighbor selection? This could further clarify the value and limitations of using a learned router.

**Ethical Concerns:**

["NO or VERY MINOR ethics concerns only"]

**Final Justification:**

The rebuttal has addressed most of my concerns. While the proposed method introduces additional overhead, the authors have presented an alternative inference approach that maintains similar task performance with reduced overhead. Thus I decide to increase my scores. Additionally, I hope the authors can further improve the presentation and the clarity of their method in the revised manuscript.

**Limitations:**

The authors have discussed limitations.

**Quality:**

3

**Strengths And Weaknesses:**

Strengths:
- The idea of introducing switchable codebooks at the image-level and token-level to improve quantization capacity is well motivated.
- The proposed training strategy appears compatible with standard latent-space models, making the approach plug-and-play.

Weaknesses:
- While STSCQ maintains a similar bit allocation to global-codebook methods (with negligible overhead for routing), the decoder must store and access a large number of codebooks at inference time (up to $M \times T$). This additional memory/storage overhead is not reported or discussed.
- The ablation study in Section A.4 shows performance degradation when increasing the number of routing codebooks, suggesting potential limitations of the method on more complex or diverse datasets.
- The paper’ presentation is not clear enough. See detailed questions below.

---

> ### Author Rebuttal · Authors · 2025-07-31
>
> **Answer 1:**
>
> We have provided a detailed analysis of the required storage overhead and inference latency on a V100 GPU, as shown in the table below. Admittedly, selecting the codebook group based on the minimum error during inference does introduce additional overhead, which is particularly noticeable when using MaskGIT-VQGAN with 256 tokens as the backbone. To address this, we also experimented with directly utilizing the routing results during inference. In this approach, the model only needs to load the codebook group indicated by the routing decision, rather than maintaining all codebook groups on the decoder side. This strategy offers clear advantages, significantly reducing both storage overhead and inference latency. In the future, we also plan to further mitigate this issue through optimization techniques such as model quantization or pruning.
>
> **Table: Inference latency and memory overhead statistics. For ours method, 'NN' and 'CR' refer to Nearest-Neighbor search and Codebook Routing search respectively.**
>
> |Backbone|bpp|Token-specific|Codebooks|Infer Time (s / img)|GPU Mem (MB)|Storage (MB)|FLOPs|
> |-|-|-|-|-|-|-|-|
> |Maskgit-VQGAN|0.0469|-|1|0.1495|474.14|94.69|158.14 G|
> |Ours (NN)|0.0469|✓|1|0.1554|3283.39|1122.70|158.14 G|
> |Ours (NN)|0.0391|✓|16|0.1771|12508.17|4194.70|158.14 G|
> |Ours (CR)|0.0391|✓|16|0.1544|5116.15|354.70|158.14 G|
> |TiTok-S|0.0234|-|1|0.1433|586.03|98.68|113.57 G|
> |Ours (NN)|0.0234|✓|1|0.1437|609.84|122.68|113.57 G|
> |Ours (NN)|0.0157|✓|256|0.1496|1894.54|482.71|113.57 G|
> |Ours (CR)|0.0157|✓|256|0.1450|1157.62|100.21|113.57 G|
> |TiTok-L|0.0059|-|1|0.1728|2709.24|1157.61|269.62 G|
> |Ours (NN)|0.0059|✓|1|0.1744|2716.80|1163.47|269.62 G|
> |Ours (NN)|0.0040|✓|256|0.1750|2865.43|1253.50|269.62 G|
> |Ours (CR)|0.0040|✓|256|0.1741|2804.02|1157.87|269.62 G|
>
> **Answer 2:**
>
> Our method can be readily transferred to images from other domains and is equally applicable to more complex and diverse datasets. For example, we report experimental results on ImageNet-1K, which contains more than 1M images from different categories. As shown in the table below, our approach achieves significant improvements over the original baseline across various metrics. Specifically, the FID score is reduced from 3.91 to 2.27, and the LPIPS score is improved from 0.3889 to 0.3564.
>
> **Table: Evaluation of reconstruction performance on the ImageNet-1K 256 × 256.**
>
> |Method|bpp|FID ↓|LPIPS ↓|SSIM ↑|PSNR ↑|
> |-|-|-|-|-|-|
> |TiTok-S|0.0234|3.91|0.3889|17.48|0.4034|
> |Ours|0.0234|2.27|0.3564|17.67|0.4139|
>
> As discussed in Section A.4 of the supplementary material, we did observe that when increasing the number of routing groups while keeping the size of each codebook fixed, the evaluation metrics first improved and then declined. We attribute this to the limited size of our training dataset CASIA-WebFace, which is insufficient to fully utilize the large codebook space. Additionally, increasing the number of groups raises the routing complexity, resulting in under-trained embeddings within each codebook group. These factors together negatively impact the evaluation metrics. We plan to address this issue in our future work.
>
> **Answer for Q1:**
>
> We apologize for the oversight—there is an error in Eq. 8 in the manuscript. The correct form is $\mathcal{L}\_{\text{ent}} = \sum\_{i=1}^M \bar{g\_{\theta}}(\mathbf{z}\_e) \log \bar{g\_{\theta}}(\mathbf{z}\_e)$, where $\bar{g\_{\theta}}(\mathbf{z}\_e)$ denotes the average probability of the current codebook group across samples in the batch. This loss is to encourage the router to assign different codebook groups to samples within the same batch, thereby making full use of the expanded codebook latent space.
>
> **Answer for Q2:**
>
> We appreciate you highlighting this oversight. You are correct—this was an editorial error, and the second $C_1$ should indeed be $C_2$. Regarding the codebook quantization in our method, when both switchable codebook quantization and token-specific codebook quantization are used in combination, the total number of sub-codebooks is indeed $M \times T$. Furthermore, $C_{global}$ refers to the globally shared codebook in previous methods, where all $T$ tokens share a common set of $K$ embeddings. We hope this clarification addresses your concern.
>
> **Answer for Q3:**
>
> We investigated the utilization of sub-codebooks when assigning separate codebooks to individual tokens. Our findings reveal that these sub-codebooks exhibit no sign of redundancy. Conversely, the token-specific codebook strategy not only further enhances the codebook utilization per token but also mitigates the imbalanced utilization allocation observed when using a single, shared codebook across all tokens, as shown in the table below. For those tokens with low utilization, we believe this may be due to their limited information content, which is related to the performance of the encoder. In the future, it would be reasonable to assign codebooks of varying sizes to different tokens to further reduce redundancy.
>
> **Table: Codebook utilization rates (%) per token on LFW dataset.**
>
> |Method|bpp|Min|Max|Mean|STD|
> |-|-|-|-|-|-|
> |Global-shared|0.0234|3.49|78.12|54.17|14.71|
> |Ours|0.0234|17.90|83.89|74.02|9.14|
>
> **Answer for Q4:**
>
> We have now supplemented the main results in Table 1 by adding the corresponding number of codebook groups ($M$) and the size of individual codebooks ($K$). The updated results are shown below.
>
> It is important to note that the main results presented here solely utilize the token-specific strategy. Consequently, the switchable codebook strategy—specifically designed to reduce the bits per pixel (bpp)—is not included in this main results table. Instead, it can be found in Table 2 within the main body of the paper.
>
> **Table: Detailed quantitative comparison with state-of-the-art methods.**
>
> | Method| Model Type | # Tokens | # Codebooks | Codebook Size | MeanAcc(%) | IDS| bpp|
> |-|-|-|-|-|-|-|-|
> | MaskGit-VQGAN| 2D| 256| 1| 4096| 90.70| 0.6314 | 0.0469 |
> | TiTok-S| 1D| 128| 1| 4096| 87.56| 0.5764 | 0.0234 |
> | TiTok-L| 1D| 32| 1| 4096| 65.07| 0.1812 | 0.0059 |
> | **Ours(MaskGit-VQGAN)** | 2D| 256| 1| 4096| 93.51| 0.6659 | 0.0469 |
> | **Ours(TiTok-s)**| 1D| 128| 1| 4096| 91.66| 0.6120 | 0.0234 |
> | **Ours(TiTok-l)**| 1D| 32| 1| 4096| 73.13| 0.2583 | 0.0059 |
>
> **Answer for Q5:**
>
> When quantizing each token, we employ the nearest neighbor search strategy, not the routing mechanism used at the image level. The reason for this design choice is that, if we were to continue using the routing selection mechanism for each token, it would indeed enhance the diversity of token representations. However, this approach would require storing an additional group index for every token, introducing extra overhead. This contradicts our goal of reducing bpp while increasing the overall codebook capacity. Specifically, assuming that image-level codebook routing is not considered, an image is represented by 256 tokens. If each token-specific codebook is divided into $M$ groups, each containing $K$ entries, then the number of bits required to represent an image is $256 \times (\log M + \log K)$. For example, given an original codebook size of 4096, the required storage is $256 \times \log 4096 = 3072$ bits. However, if we partition the codebook into 256 groups, each of size 256, the storage increases to $256 \times (\log 256 + \log 256) = 4096$ bits.
>
> **Answer for Q6:**
>
> Yes, we have compared the codebook utilization between nearest neighbor search and our routing selection mechanism, and the results are shown in the table below. When using nearest neighbor search, both codebook utilization and evaluation metrics drop significantly—especially codebook utilization, which falls to only 19.89\%. This clearly contradicts our original motivation for introducing multiple codebook groups. We believe this is because, with nearest neighbor search, the model tends to favor codebooks that have been more extensively optimized, leading to redundancy in other codebooks. Therefore, we introduce the routing selection mechanism to encourage the model to make full use of as many codebooks as possible, thereby leveraging the diversity of the codebook latent space.
>
> **Table: Comparisons of different codebook selection strategies used while training.**
>
> | Method| Codebook Selection | # Tokens | # Codebooks | Codebook Size | MeanAcc(%) | IDS| Codebook Mean Utilization(%) |
> |-|-|-|-|-|-|-|-|
> | Ours(TiTok-S)  | NN | 128| 16| 1024| 89.28| 0.5701 | 19.89|
> | Ours(TiTok-S)  | Routing| 128| 16| 1024| 89.89| 0.5740 | 89.34|

---

> > ### Comment · Reviewer_whZv · 2025-08-04
> >
> > Thank you for the rebuttal, which has addressed most of my concerns. While the proposed method introduces additional overhead, the authors have presented an alternative inference approach that maintains similar task performance with reduced overhead. I encourage the authors to further improve the presentation in the revised manuscript. I am increasing my scores accordingly.

---

### Official Review · Reviewer_bPDd · 2025-07-04

**Clarity:** 3
**Significance:** 3
**Originality:** 2
**Rating:** 4
**Confidence:** 4

**Summary:**

This paper proposes a switchable token-specific codebook quantization framework for face image compression. To address the capacity limitations of conventional codebook-based image compression methods, it reorganizes the codebook at multiple levels to decompose the problem into smaller subproblems by integrating both an image-level switchable codebook and a token-level sub-codebook quantization mechanism. To enable effective training of this hierarchical structure, the authors further design a novel three-stage training strategy. The idea has practical significance and, as one of the first attempts to realize such a concept, demonstrates a clear degree of originality.

**Questions:**

a) However, the paper has several weaknesses. It does not include visual examples of the reconstructed compressed images, so the perceptual quality of the images is hard to assess. Furthermore, it remains unclear whether decreasing the single codebook size might damage the ability to preserve token diversity, thus potentially leading to information loss.

b) Also, the MoE-based routing design may lead to multiple codebooks learning redundant representations, but the paper does not provide explicit evidence to ensure codebook diversity. Additionally, the inference latency and memory overhead introduced by the multiple routing paths and token-specific codebooks are not analyzed, nor is the training resource requirement compared to other baselines.

c) The method has only been validated on face image datasets, so it remains unclear whether the token-specific strategy would generalize well to other domains such as natural scenes or anime images, where the data distribution could be more complex. In addition, the approach heavily relies on the quality of the pretrained autoencoder, and the impact of different encoder-decoder backbones has not been explored. It would also be helpful to clarify how well this design could transfer to real-world edge devices with limited computational or memory resources.

**Ethical Concerns:**

["NO or VERY MINOR ethics concerns only"]

**Final Justification:**

The reviewer went through the discussion and made the final remarks. This is the final decision.

**Limitations:**

yes

**Quality:**

3

**Strengths And Weaknesses:**

The concept of using both image-level and token-level codebook quantization mechanisms is novel and timely, and the joint optimization approach is new in the context of codebook compression. During training, the integration of the MoE strategy and the loss functions is well designed. Moreover, with extensive benchmarks and detailed ablation studies, the experimental results are comprehensive and convincing. The paper is well written with clear figures and formulas. Finally, the proposed approach provides a novel framework for ultra-low bitrate face image compression, which can serve as a valuable resource for future research.

---

> ### Author Rebuttal · Authors · 2025-07-31
>
> **Answer (a):**
>
> Thank you for your valuable suggestion. To better assess perceptual quality, we have included visualization examples in the supplementary material. However, due to policy limitations, we regret that we are unable to incorporate additional visual examples at this stage. To further validate the image generation quality of our method, we have conducted quantitative evaluations of image quality metrics, as presented in the table below. We are committed to providing more comprehensive visualizations in future iterations to further demonstrate the efficacy of our approach.
>
> **Table: Quantitative comparison about image quality with baseline methods.**
> |Method|Model Type|Tokens|bpp|SSIM ↑|PSNR ↑|
> |-|-|-|-|-|-|
> |TiTok-S|1D|128|0.0234|0.7799|27.21|
> |TiTok-L|1D|32|0.0059|0.6868|23.25|
> |Ours(TiTok-S)|1D|128|0.0234|0.7823|27.39|
> |Ours(TiTok-L)|1D|32|0.0059|0.6931|23.33|
>
> Regarding "decreasing the single codebook size might damage the ability to preserve token diversity", in fact, directly reducing the size of a single codebook can indeed weaken its representational capacity. However, our method addresses this issue in two perspectives.
>
> First, we argue that a more diverse or larger codebook feature space is not always optimistic solution; rather, it should be well-matched to the latent space of the encoder outputs. If the codebook covers a very broad space but the tokens cannot effectively utilize it, this will lead to low codebook utilization and wasted resources.
>
> Second, We observed that the original strategy of using a global-shared codebook across all tokens leads to an uneven distribution of codebook utilization. As shown in the table below, different tokens exploit the shared codebook but with varying degrees. To address this, we propose a token-specific codebook strategy, where each token learns its own codebook to maximize the preservation of its visual characteristics. With our approach, the issue of uneven codebook utilization among tokens is significantly alleviated. This demonstrates that our codebook is better able to fit the feature space of the encoder outputs. As a result, the feature information of each token can be more effectively quantized by the current codebook, thereby reducing quantization errors caused by codebook utilization imbalance.
>
> **Table: Codebook utilization rates (%) per token on LFW dataset.**
>
> |Method|bpp|Min|Max|Mean|STD|
> |-|-|-|-|-|-|
> |Global-shared|0.0234|3.49|78.12|54.17|14.71|
> |Ours|0.0234|17.90|83.89|74.02|9.14|
>
> Finally, by introducing multiple groups at the image level and employing a routing strategy to assign the most suitable group to each image, our approach effectively increases the overall codebook capacity, even if the size of each individual codebook is reduced. Moreover, each codebook becomes more specialized in capturing the common features of a particular category of images, which helps maintain the diversity of the codebook representation space.
>
> **Table: Total codebook utilization rates (%) under five face recognition datasets.**
>
> |bpp|Tokens|Codebooks|Codebook Size|LFW|CPLFW|CFPFP|AgeDB|CALFW|Mean Utilization|
> |-|-|-|-|-|-|-|-|-|-|
> |0.0234|128|1|4096|74.01|74.24|60.57|61.44|67.07|**67.47**|
> |0.0196|128|16|1024|90.39|94.68|87.53|86.19|87.93|**89.34**|
> |0.0157|128|256|256|93.15|96.45|94.46|91.89|92.00|**93.59**|
>
> **Answer (b):**
>
> To maintain the representational capacity of the codebook latent space as individual codebook size decreases, our method employs multiple groups of codebooks and effectively utilizes them.
> To better evaluate the utilization of these multiple codebook groups, we used the TiTok-S baseline as an example. We tested their codebook utilization under three different bpps, and the results are presented in the table below.
>
>
> As shown in the table, the overall codebook utilization increases as the number of codebook groups grows. This result supports our hypothesis that, by organizing codebooks into groups, we can use smaller codebooks to capture more representative features. Consequently, this approach allows us to maintain the diversity of the codebook representation space while reducing the bpp during compression.
>
> We have also evaluated the inference latency and storage overhead introduced by our method on a V100 GPU, as summarized in the table below. The table reports the inference latency and required storage for three baselines under different bpp (bits per pixel) settings. As shown, the expansion of the overall codebook size does indeed incur additional storage costs, along with a slight increase in inference latency. Since the number and dimensionality of tokens in the MaskGIT baseline are both significantly higher than those in the TiTok baseline (256 vs. 128 tokens, 256 vs. 12 dimensions), the storage overhead introduced by MaskGIT is substantially greater than that of TiTok. However, we would like to emphasize that in practical applications, our primary concern is the storage and transmission overhead of the images themselves. Our goal is to minimize image storage costs while preserving image quality as much as possible. Under the same bpp, our method achieves better recognition performance and image quality. Additionally, we have also explored approaches to reduce the model storage overhead. We explored a routing-based inference: only the codebook group selected by the router needs to be loaded, rather than searching with the minimum error across all codebooks. With this improvement, both inference latency and storage overhead are substantially alleviated, while the performance remains competitive with the baselines. There are also several engineering approaches to mitigate storage overhead. For example, we suggest performing quantization on the encoding side so that both the features and the codebook are matched within the int8 range. After obtaining the index sequence, the original fp32 codebook can be used on the cloud side for reconstruction and decoding.
>
> **Table: Inference latency and memory overhead statistics. For ours method, 'NN' and 'CR' refer to Nearest-Neighbor search and Codebook Routing search respectively.**
>
> |Backbone|bpp|Token-specific|Codebooks|Infer Time (s / img)|GPU Mem (MB)|Storage (MB)|FLOPs|
> |-|-|-|-|-|-|-|-|
> |Maskgit-VQGAN|0.0469|-|1|0.1495|474.14|94.69|158.14 G|
> |Ours (NN)|0.0469|✓|1|0.1554|3283.39|1122.70|158.14 G|
> |Ours (NN)|0.0391|✓|16|0.1771|12508.17|4194.70|158.14 G|
> |Ours (CR)|0.0391|✓|16|0.1544|5116.15|354.70|158.14 G|
> |TiTok-S|0.0234|-|1|0.1433|586.03|98.68|113.57 G|
> |Ours (NN)|0.0234|✓|1|0.1437|609.84|122.68|113.57 G|
> |Ours (NN)|0.0157|✓|256|0.1496|1894.54|482.71|113.57 G|
> |Ours (CR)|0.0157|✓|256|0.1450|1157.62|100.21|113.57 G|
> |TiTok-L|0.0059|-|1|0.1728|2709.24|1157.61|269.62 G|
> |Ours (NN)|0.0059|✓|1|0.1744|2716.80|1163.47|269.62 G|
> |Ours (NN)|0.0040|✓|256|0.1750|2865.43|1253.50|269.62 G|
> |Ours (CR)|0.0040|✓|256|0.1741|2804.02|1157.87|269.62 G|
>
> **Table: Quantitative comparison between different codebook selection policies during inference.**
>
> |Backbone|Codebook Selection|bpp|MeanAcc(%)|IDS|
> |-|-|-|-|-|
> |Maskgit-VQGAN|NN|0.0391|92.23|0.6264|
> |Maskgit-VQGAN|CR|0.0391|92.22|0.6253|
> |TiTok-S|NN|0.0157|87.60|0.5180|
> |TiTok-S|CR|0.0157|87.54|0.5125|
> |TiTok-L|NN|0.0040|66.02|0.1885|
> |TiTok-L|CR|0.0040|65.65|0.1864|
>
> Regarding the comparison of training resources, the original paper reports that training TiTok-S requires 32 A100-40G GPUs for 50 hours. In contrast, our method primarily focuses on training the codebook, while the encoder is directly adopted from pretrained weights and the decoder is correspondingly fine-tuned. As a result, our training cost is 8 V100 GPUs for nearly 40 hours. We hope this clarification helps you better understand the differences in training resource requirements.
>
> **Answer (c)**
>
> Our proposed token-specific strategy can be easily adapted to images from other domains as well. To better evaluate the generalizability of our method, we also conducted experiments on the larger and more diverse ImageNet-1K dataset, with the results presented in the table below. As shown, our approach achieves significant improvements over the original baseline across various metrics. For example, the FID score is reduced from 3.91 to 2.27, and the LPIPS score is improved from 0.3889 to 0.3564. These results demonstrate the necessity and effectiveness of learning a separate codebook for each token.
>
> **Table: Evaluation of reconstruction performance on the ImageNet-1K 256 × 256.**
>
> |Method|bpp|FID ↓|LPIPS ↓|SSIM ↑|PSNR ↑|
> |-|-|-|-|-|-|
> |TiTok-S|0.0234|3.91|0.3889|17.48|0.4034|
> |Ours|0.0234|2.27|0.3564|17.67|0.4139|
>
> As you mentioned, our method relies on the performance of the pre-trained autoencoder baseline, but it can be conveniently integrated into any existing codebook-based image compression framework. Here, we present updated results based on the IBQ baseline published at ICCV 2025 [1], where our approach also demonstrates its effectiveness, as shown in the table below.
>
> **Table: Evaluation on state-of-the-art IBQ.**
>
> |Method|Model Type|Tokens|MeanAcc(%)|IDS|bpp|
> |-|-|-|-|-|-|
> |IBQ|2D|256|95.73|0.8403|0.0508|
> |Ours(IBQ)|2D|256|96.13|0.8545|0.0508|
>
> The goal of image compression is to balance the trade-off between compression rate and distortion, aiming to minimize storage costs while preserving as much information from the original image as possible. In practical applications, we recommend deploying a lightweight encoder on edge devices to obtain high-quality compressed representations at lower bpp using our method, which can then be decoded or further processed by a decoder deployed in the cloud. Since our approach remains effective even at extremely low bpp (<0.01), it can significantly reduce the cost of image transmission and storage.
>
> Reference: [1] Shi F, Luo Z, Ge Y, et al. Taming scalable visual tokenizer for autoregressive image generation. ICCV 2025.

---

### Note · Authors · 2025-08-15

We thank the reviewers, ACs, and SACs for their valuable suggestions. As we have discussed, our proposed method introduces a routing-based codebook selection mechanism, in which the router is trained with entropy regularization to ensure efficient codebook utilization and to learn codebook representations that are specific to different facial attributes. Additionally, our token-specific codebook design further enhances the diversity of token representations without increasing the overall bpp, and alleviates the previous problem of imbalance in codebook usage across different tokens when using a single shared codebook. Our approach not only enables flexible bpp adjustment, but also maximizes codebook diversity and maintains performance even at lower bpp values.

To address the issue of inference cost, we explored employing the routing results directly during inference. The experimental results demonstrate that our method can substantially reduce storage requirements while maintaining comparable performance; for example, on MaskGIT, we achieve only a 0.01% drop in performance with about 8% of the original storage cost, and achieved a per-image inference speedup of 0.02 seconds. This highlights the practical feasibility of our method in real-world applications.

Furthermore, we have also conducted additional experiments on more baselines and datasets. For example, on the latest SOTA IBQ baseline, our method improved the MeanAcc from 95.73 to 96.13, and increased the IDS from 0.8403 to 0.8545. On the more diverse ImageNet dataset, our approach improved the FID from 3.91 to 2.27 on the TiTok-S baseline. These results demonstrate that our method can be effectively transferred to more complex and diverse scenarios while consistently maintaining its effectiveness.

Moreover, although we are unable to provide additional qualitative visualizations, we present quantitative comparisons of image quality metrics. For example, on the TiTok-S baseline, our method improves SSIM from 0.7799 to 0.7823 and PSNR from 27.21 to 27.39, demonstrating our advantage in terms of image quality.

We will provide further updates and more results in future versions.

---

### Decision · Program_Chairs · 2025-09-17

**Decision:**

Accept (poster)

**Comment:**

In this paper the aurhors introduce a switchable, token-specific codebook quantization framework for face image compression. Unlike conventional codebook-based methods, which are often constrained by limited capacity, the proposed approach restructures the codebook at multiple levels. It combines an image-level switchable codebook with a token-level sub-codebook quantization mechanism. This breaks down the overall task into smaller, more manageable subproblems. To effectively train this hierarchical design, the authors devise a novel three-stage training strategy. As one of the first implementations of this idea, the method shows both practical significance and a notable degree of originality.
Strenghts include (i) novel idea of combining image-level and token-level codebook, (ii) novel idea of joint optimization approach for this problem, (iii) solid experimental evaluation, (iv) sensible ablation study, (v) good results, (vi) well written.
Weaknesses include (i) lack of visual illustrations, (ii) only tested on face images, (iii) risk of redundant representations, (iv) high memory requirements for the decoder.
The paper got mixed reviews before the rebuttal, but after the rebuttal and the following thorough discussions, the reviewers where predominantly positive with scores 4,4,3,5.